# Inferring Hybrid Neural Fluid Fields from Videos

**Hong-Xing Yu**[1*]    **Yang Zheng**[1*]    **Yuan Gao**[1]    **Yitong Deng**[1]    **Bo Zhu**[2]    **Jiajun Wu**[1]

Stanford University[1]                Georgia Institute of Technology[2]

## Abstract

We study recovering fluid density and velocity from sparse multiview videos. Existing neural dynamic reconstruction methods predominantly rely on optical flows; therefore, they cannot accurately estimate the density and uncover the underlying velocity due to the inherent visual ambiguities of fluid velocity, as fluids are often shapeless and lack stable visual features. The challenge is further pronounced by the turbulent nature of fluid flows, which calls for properly designed fluid velocity representations. To address these challenges, we propose hybrid neural fluid fields (HyFluid), a neural approach to jointly infer fluid density and velocity fields. Specifically, to deal with visual ambiguities of fluid velocity, we introduce a set of physics-based losses that enforce inferring a physically plausible velocity field, which is divergence-free and drives the transport of density. To deal with the turbulent nature of fluid velocity, we design a hybrid neural velocity representation that includes a base neural velocity field that captures most irrotational energy and a vortex particle-based velocity that models residual turbulent velocity. We show that our method enables recovering vortical flow details. Our approach opens up possibilities for various learning and reconstruction applications centered around 3D incompressible flow, including fluid re-simulation and editing, future prediction, and neural dynamic scene composition. Project website: https://kovenyu.com/HyFluid/.

## 1  Introduction

Fluid is ubiquitous in our surroundings, from a small breeze in the morning to large-scale atmosphere flow that would affect the weather in the following week. Understanding and predicting fluid dynamics play a central role in climate forecasting (Bauer et al., 2015), vehicle design (Bushnell & Moore, 1991), visual special effect (Pfaff et al., 2010), etc. Yet, it remains an open problem in scientific machine learning to accurately recover fluid flows from visual observations. Flow motions can only be seen indirectly, and, unlike solids that have certain shapes and simple constrained motion patterns, fluid systems have intricate and complex dynamics that exhibit different features across spatial scales. These characteristics pose unique challenges in the representations and algorithms to recover physically correct velocity. Recently, physics-informed neural networks (Raissi et al., 2019) have shown promise in recovering velocity fields from density fields for scientific computing. These neural methods are scalable and flexible compared to traditional grid-based methods, yet they require accurate density fields that are unavailable in uncontrolled ordinary scenes.

In this work, we focus on recovering the density[†] and velocity of fluids from sparsely captured multiview videos (see Fig. 1 for illustration). We identify three key challenges. Firstly, the fluid velocity is ambiguous from visual observations. Unlike solid objects characterized by consistent

---

[*]Equal contributions.

[†]"Density" refers to the concentration of the fluid substance such as smoke soot. Not to be confused with the physical density, i.e., mass over volume.

37th Conference on Neural Information Processing Systems (NeurIPS 2023).

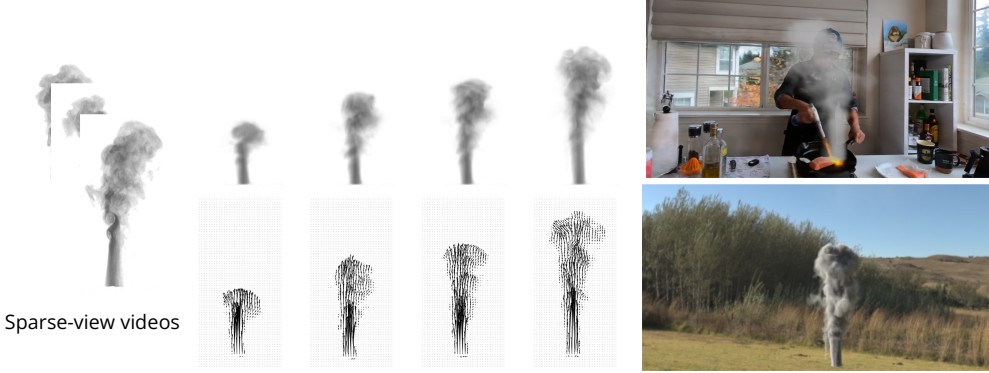

Figure 1: We aim at inferring hybrid neural fluid representations of density and velocity from sparse multi-view videos. This allows novel view video synthesis, re-simulation, and dynamic scene composition.

shapes and stable visual features, fluids are inherently shapeless, often monochromatic, and semi-transparent. This lack of physical definition makes visual tracking an implausible task, contributing to pronounced visual ambiguity, particularly in regions of laminar flow where the fluid appearance hardly changes. This challenge is beyond the capacity of general-purpose neural dynamic reconstruction methods (Li et al., 2021; Du et al., 2021), as they predominantly depend on optical flow which is ineffective in this context.

Secondly, accurately depicting fluid velocity necessitates appropriate representations that respect its turbulent features. Fluid flows exhibit varying features across multiple spatial scales (Frisch & Donnelly, 1996), a phenomenon that is challenging to capture within a single neural representation due to its spectral bias (Rahaman et al., 2019). At the human scale, fluid motions express not only laminar flows but also turbulence (Pope, 2000). This results in a complicated entanglement of rotational, shearing, and smooth motions. The turbulent flows are high-frequency in both space and time. Therefore, the representation of such flow should be able to accommodate a broad spectrum of signal frequencies while also preserving the unique structure of rotational flows.

Lastly, it is inherently ambiguous to reconstruct physically plausible 3D density fields from sparse observations. Recovering fluid density and appearance from limited 2D videos is ill-posed primarily due to the intricate non-linear absorption and scattering processes associated with semi-transparent fluids (Max, 1995). Consequently, the reconstruction of under-constrained continuous density fields from sparse observations is an intrinsically difficult task.

To address these challenges, we propose hybrid neural fluid fields (HyFluid), a neural approach to jointly infer fluid density and velocity from sparse-view videos. At the core of HyFluid are two key technical contributions including a set of simple yet effective physics-based losses and a hybrid neural velocity representation. Our physics-based losses leverage physical constraints from the Navier-Stokes equations to jointly learn physically plausible fluid density and velocity fields. The physics-based losses enforce inferring a divergence-free velocity field that drives the transport of the density field. In addition to the new losses, we propose a hybrid fluid velocity representation as it is difficult for a single neural representation to capture turbulent fluid velocity. The main idea is to decompose our fluid velocity field into a base and a residual turbulent velocity field. The base velocity field is represented by a neural field that captures most irrotational energy, and the residual velocity field is represented by vortex particles that feature highly rotational and shearing velocities. In order to address the 2D-to-3D density ambiguity, we leverage both visual imaging signals by differentiable volume rendering and the physical fluid transport constraint to regularize the density field.

We demonstrate that HyFluid can yield high-fidelity recovery of density and velocity, allowing novel view video synthesis, re-simulation, editing, future prediction, and neural dynamic scene composition (Fig. 1). In summary, our contributions are three-folded: (1) We propose hybrid neural fluid fields (HyFluid), a neural approach to infer fluid density and velocity from sparse multiview videos. (2) We show that using simple physics-based losses and a hybrid neural velocity representation allows uncovering turbulent fluid motion from sparse observations. (3) We evaluate our method on novel-

view re-simulation and novel-view future prediction to benchmark joint reconstruction for real fluids, and we show that our approach delivers high-fidelity synthesis and reconstructions in comparison to state-of-the-art neural methods.

## 2 Preliminaries

**Incompressible Navier-Stokes equations.** Our motion representation and learning signals are motivated by incompressible Navier-Stokes equations. With a low flow speed, fluid motion is considered incompressible and can be well described by:

$$\frac{D\mathbf{u}}{Dt} = \frac{\partial \mathbf{u}}{\partial t} + \mathbf{u} \cdot \nabla \mathbf{u} = -\frac{1}{\rho}\nabla p + \nu\nabla \cdot \nabla \mathbf{u} + \mathbf{f}, \tag{1}$$

$$\nabla \cdot \mathbf{u} = 0. \tag{2}$$

Here, Eqn. 1 is known as the momentum equation, where the LHS is the material derivative for velocity, $\frac{D\mathbf{u}}{Dt} = \frac{\partial \mathbf{u}}{\partial t} + \mathbf{u} \cdot \nabla \mathbf{u}$, which represents the time rate of change of velocity $\mathbf{u}$ of a fluid parcel, and the RHS is the momentum induced by pressure gradient $-\frac{1}{\rho}\nabla p$ (where $\rho$ denotes the physical density of the fluid parcel), viscosity $\nu\nabla \cdot \nabla \mathbf{u}$, and some external force $\mathbf{f}$. We consider inviscid fluid and thus drop the viscosity term. Eqn. 2 is known as the mass conservation equation, meaning that the mass flowing into a controlled volume should be equal to the mass flowing out of the volume.

**Differentiable volume rendering.** We integrate volume rendering for joint learning of visual appearance, density, and velocity from videos. Volume rendering is a suitable model for translucent materials and participating media like smoke and fog (Kajiya & Von Herzen, 1984). In volume rendering, the radiance $L(\mathbf{o}, \mathbf{d})$ arriving at the camera location $\mathbf{o}$ from direction $\mathbf{d}$ (a.k.a. the radiance of a camera ray $\mathbf{r}(t) = \mathbf{o} - t\mathbf{d}$) with near and far bounds $t_n$ and $t_f$ is given by

$$L(\mathbf{o}, \mathbf{d}) = \int_{t_n}^{t_f} T(t)\sigma(\mathbf{r}(t))L_{\mathrm{e}}(\mathbf{r}(t), \mathbf{d})dt, \tag{3}$$

where $T(t) = \exp(-\int_{t_n}^{t} \sigma(\mathbf{r}(s))ds)$ denotes the transmittance along the ray from $t_n$ to $t$, $L_{\mathrm{e}}$ denotes the emitting radiance, and $\sigma$ denotes the optical density which can be considered proportional to the concentration density of fluid substance according to Beer-Lambert law. Following recent neural rendering methods (Mildenhall et al., 2020), we use quadrature to discretize it (Max, 1995).

## 3 Hybrid Neural Fluid Fields

Our goal is to recover the fluid appearance, density, and velocity from sparse multiview RGB videos. To this end, we propose hybrid neural fluid fields (HyFluid). At the core of HyFluid are a set of simple physics-based losses and a hybrid neural velocity representation. In particular, HyFluid aims to infer a neural density field $\sigma(x, y, z, t)$, the appearance of the fluid $L_{\mathrm{e}}$, and the underlying velocity $\mathbf{u}(x, y, z, t) = \mathbf{u}_{\mathrm{base}}(x, y, z, t) + \mathbf{u}_{\mathrm{vort}}(x, y, z, t)$, where we decompose it into a base neural velocity field $\mathbf{u}_{\mathrm{base}}$ and a residual vortex particle-driven turbulent velocity. $\mathbf{u} = [u, v, w]$ denotes the velocity along $x, y, z$ axis, respectively. In the following, we first introduce the physics-based losses, then we describe our hybrid velocity representation, and finally, we summarize our learning signals for the joint inference. We show a conceptual illustration in Fig. 2.

**Physics-based losses.** We propose physics-based losses $\mathcal{L}_{\mathrm{physics}}$ including a density loss, a projection loss, and a laminar regularization loss.

*Density loss.* While the neural density field can be learned through differentiable volume rendering, the underlying velocity field is hidden from the visual observations. Unlike solids that have regular shapes and invariant visual features to track, fluids are often shapeless, monochromatic, and semi-transparent. This makes it implausible to visually track it. To uncover the hidden velocity from visual observations, we resort to the density transport equation in incompressible flows, $\frac{D\sigma}{Dt} = 0$, and introduce a physics-informed supervision signal:

$$\mathcal{L}_{\mathrm{density}} = \underset{x,y,z,t}{\mathbb{E}} \left[ \frac{\partial \sigma}{\partial t} + u \cdot \frac{\partial \sigma}{\partial x} + v \cdot \frac{\partial \sigma}{\partial y} + w \cdot \frac{\partial \sigma}{\partial z} \right]. \tag{4}$$

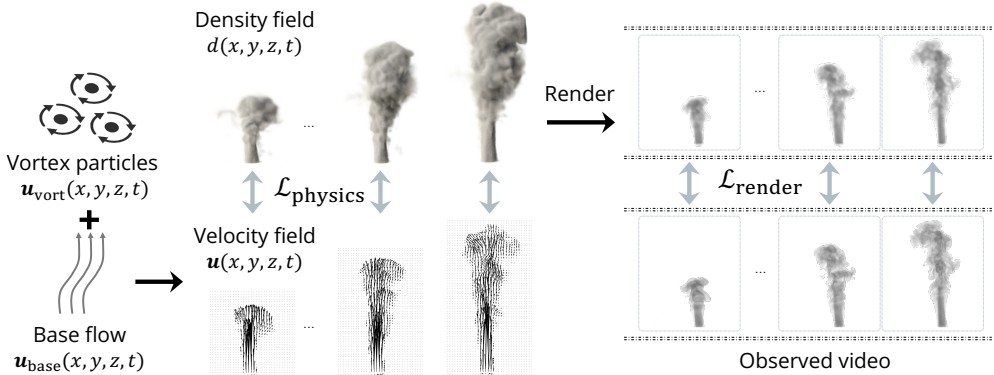

Figure 2: Illustration of hybrid neural fluid fields (HyFluid) which aims to jointly infer fluid density and velocity from videos using visual signals via differentiable volume rendering and physics-based losses. To facilitate learning turbulent velocity, we decompose the velocity field into a base velocity and a vortex particle-driven velocity. See the main text for detailed descriptions.

This loss says that the 3D velocity field $(u, v, w)$ at any given moment $t$ should transport the density field $\sigma$ such that it evolves to the density field at the next moment, in a similar spirit to scene flow (Vedula et al., 1999).

*Projection loss.* Another important physical property of incompressible flow is mass conservation depicted in Eqn. 2, that is, the velocity field should be divergence-free. A straightforward way to enforce divergence-free condition is to impose a loss on the divergence of velocity, $\frac{\partial u}{\partial x} + \frac{\partial v}{\partial y} + \frac{\partial w}{\partial z}$. However, we find that this empirically leads to a degenerated solution, where only the laminar flows (i.e., smooth, direct-current flows) can be uncovered as this is a trivial local minimum for every point. Yet, the incompressibility of fluid flows is confined through a global dynamic system, modulated by the pressure of the fluid.

Therefore, we need a globally-aware, physically-plausible supervision that does not conform to trivial local minima. Motivated by the commonly used pressure projection solver in computational fluid dynamics, we propose a projection loss that constrains our learned velocity to be divergence-free:

$$\mathcal{L}_{\text{proj}} = \mathop{\mathbb{E}}_{x,y,z,t} \left[ \|\mathbf{u} - \mathbf{u}_p\|^2 \right], \tag{5}$$

where $\mathbf{u}_p = \texttt{project}(\mathbf{u})$ are the pressure-projected velocity field, using a pressure `project` solver. A `project` solver implements a Helmholtz decomposition by solving a global linear system for pressure, and it projects the velocity field $\mathbf{u}$ to a divergence-free manifold using the gradient of the solved pressure. Also, we notice that any divergence-free velocity field $\mathbf{u}$ incurs zero projection loss as it is an identity mapping for divergence-free fields. Thus, this projection loss essentially constrains the uncovered velocity within a divergence-free subspace, and it is not subject to trivial local minima of pure direct-current flows.

*Laminar regularization loss.* These density and projection losses do not warrant physically-correct velocity reconstruction for laminar flows. In laminar flows, all partial derivatives of density are zero, so any velocity, e.g., all-zero velocity, incurs zero $\mathcal{L}_{\text{density}}$. As for $\mathcal{L}_{\text{project}}$, all-zero velocity fields also incur a zero loss. Hence, we introduce a laminar regularization loss:

$$\mathcal{L}_{\text{laminar}} = \mathop{\mathbb{E}}_{x,y,z,t} \left[ \max(0, \gamma\sigma - \|\mathbf{u}\|) \right], \tag{6}$$

which is a hinge loss that encourages high-density regions to have non-zero velocity, and $\gamma$ denotes a hyperparameter to scale the threshold according to the velocity magnitude unit. These three losses are complementary and interconnected to uncover a physically plausible fluid flow.

**Hybrid neural velocity representation.** At the human scale, fluid flows are often turbulent. They exhibit complex dynamic features including circular and shearing flows. In addition to the common challenges of reconstructing non-turbulent flows, turbulent flows are even more intricate and difficult to model and capture. They are high-frequency flows in both space and time. This demands that the velocity representation be able to accommodate a wide range of signal frequencies and the particular

structure of rotational flows. However, neural representations are known to have certain spectral inductive biases toward representing low-frequency signals (Rahaman et al., 2019). While this can be alleviated by position embedding (Tancik et al., 2020) or sinusoidal activation functions (Sitzmann et al., 2020), they still lack a proper structure to capture and represent highly detailed turbulent flows.

We propose a hybrid neural velocity representation to tackle this challenge. The main idea is to decompose the underlying flow field $\mathbf{u} = \mathbf{u}_{\text{base}} + \mathbf{u}_{\text{vort}}$ into a base neural velocity field $\mathbf{u}_{\text{base}}$ and a residual vorticity-driven velocity $\mathbf{u}_{\text{vort}}$. The base neural velocity field captures the large-scale flow motion, and the vorticity-driven velocity is a dedicated representation of the small-scale vortex details that are hard to characterize by the base field. For the base velocity field $\mathbf{u}_{\text{base}}$, we extend instance neural graphics primitive (iNGP) (Müller et al., 2022) representation to the temporal domain.

The residual vorticity-driven flow $\mathbf{u}_{\text{vort}}$ is complementary to the base flow by leveraging the physical structure of turbulent fluid flows. The physical model behind the vorticity-driven flow is prescribed by the curl form of the Navier-Stokes equations (see Cottet et al. (2000) for more details):

$$\frac{D\boldsymbol{\omega}}{Dt} = \frac{\partial \boldsymbol{\omega}}{\partial t} + \mathbf{u} \cdot \nabla \boldsymbol{\omega} = \boldsymbol{\omega} \cdot \nabla \mathbf{u} + \nu \nabla \cdot \nabla \boldsymbol{\omega} + \nabla \times \mathbf{f}, \tag{7}$$

where $\boldsymbol{\omega} = \nabla \times \mathbf{u}$ denotes the vorticity. We assume inviscid fluid and conservative body force, and thus only the vortex stretching term $\boldsymbol{\omega} \cdot \nabla \mathbf{u}$ is non-zero on the RHS. In this case, Eqn. 7 becomes the vorticity transport equation, saying that vorticity is stretched and advected by the velocity flow.

This physical model has been widely used in fluid simulation (Cottet et al., 2000), known as vortex methods. In particular, we consider the vortex particle methods (Selle et al., 2005), where we use a particle-based representation for vorticity, which is low-dimensional, is easy to temporally evolve, and well embeds the circular physical flow structures in it. We represent a vortex particle $p$ at some time stamp $t$ by a triplet $\{I_p, \mathbf{x}_p^t, \boldsymbol{\omega}_p^t\}$ of its intensity $I_p$, position $\mathbf{x}_p^t$, and vorticity $\boldsymbol{\omega}_p^t$. Our vorticity-driven flow $\mathbf{u}_{\text{vort}}(\mathbf{x}, t)$ induced by vortex particles is represented by:

$$\mathbf{u}_{\text{vort}}(\mathbf{x}, t) = \sum_p I_p (\mathbf{N}_p \times \tilde{\boldsymbol{\omega}}_p), \quad \mathbf{N}_p = (\mathbf{x}_p^t - \mathbf{x})/\|\mathbf{x}_p^t - \mathbf{x}\|, \quad \tilde{\boldsymbol{\omega}}_p = K(\mathbf{x} - \mathbf{x}_p^t)\boldsymbol{\omega}_p^t, \tag{8}$$

where $K(\mathbf{x}) = \exp(-\|\mathbf{x}\|^2/2r^2)/(r^3(2\pi)^{3/2})$ is a Gaussian distribution kernel. Intuitively, every vortex particle "carries" a local circular momentum, which is itself transported with the flow prescribed by Eqn. 7. Collectively, a set of vortex particles allows learning complex flow details which are hard to capture by the base velocity field.

However, naively learning the vortex particles leads to poor local minima due to the complex interdependence of $\mathbf{u}$ and $\{I_p, \mathbf{x}_p^t, \boldsymbol{\omega}_p^t\}$. Thus, we make two simplifications. First, we assume that most energy is captured by the base flow and only use the base flow to transport vortex particles. This is achieved by first learning the base flow and then freezing the base flow to learn the residual flow. Second, we introduce a seeding strategy to disentangle learnable parameters from the particle transport: we seed overly many vortex particles on high-curl spatio-temporal locations and pre-compute the trajectory $\{\mathbf{x}_p^t\}$ and $\{\boldsymbol{\omega}_p^t\}$ for all $t$ and $p$ by the learned base flow. Therefore, the learnable parameters are reduced to $\{I_p\}$. The redundant vortex particles are automatically suppressed by learning to have zero intensities. We leave more details of the seeding strategy in the Appendix.

**Joint inference of fluid fields.** It is a highly ill-posed problem to recover density and appearance from sparse visual observations, as the imaging process for semi-transparent fluids involves complex non-linear absorption and scattering. The visual appearance of fluids depends not only on density but also lighting and fluid substance properties. Therefore, recovering turbulent, under-constrained continuous density fields from limited observations is inherently challenging.

To address this challenge, we leverage both visual imaging signals and physical fluid transport constraints by jointly learning density, appearance, and velocity. The visual signal supervision through differentiable volume rendering is given by:

$$\mathcal{L}_{\text{render}} = \mathop{\mathbb{E}}_{\mathbf{o},\mathbf{d},t} \left[ \|L_{\text{render}}(\mathbf{o}, \mathbf{d}) - L_{\text{observe}}(\mathbf{o}, \mathbf{d})\|^2 \right], \tag{9}$$

where $L_{\text{render}}$ is our volume rendered values by Eqn. 3 and $L_{\text{observe}}$ is sampled from video frames. Notice that we sample rays continuously in the viewing frustums instead of sampling through only pixel centers as NeRF-like methods do (Mildenhall et al., 2020), as we aim to learn continuous fields

and we rely on good partial derivatives from the fields. The physical transport constraint is provided by Eqn. 4, i.e., we supervise the partial derivatives of the density field. We compute all derivatives by auto-differentiation (Paszke et al., 2017). Our loss function is given by:

$$\mathcal{L} = \beta_{\text{render}}\mathcal{L}_{\text{render}} + \beta_{\text{density}}\mathcal{L}_{\text{density}} + \beta_{\text{proj}}\mathcal{L}_{\text{proj}} + \beta_{\text{laminar}}\mathcal{L}_{\text{laminar}}, \tag{10}$$

where all the $\beta$s are loss weights.

Another inherent ambiguity is between emitting radiance and density. As in Eqn. 3, the emitting radiance is multiplied by density, which means that an unconstrained spatially-varying emitting radiance field admits all sorts of trivial solutions for density. Empirically, we find little differences between using learned spatially-varying color and using a constant color. Thus, we set emitting radiance to be constant to remove this ambiguity.

## 4 Related Work

**Fluid reconstruction.** Fluid flow reconstruction from visible light measurement has been widely studied in science and engineering. Well-established methods used in controlled lab environments include active sensing techniques such as laser scanners (Hawkins et al., 2005), light path (Ji et al., 2013), and structural light (Gu et al., 2012), as well as passive marker-based methods such as particle imaging velocimetry (PIV) (Adrian & Westerweel, 2011; Elsinga et al., 2006) which injects passive particles into the fluid flows. These methods allow accurate flow measurement, yet they require specialized setup for markers, lighting, and capturing devices.

Recent methods also seek to reconstruct fluid flows from casual visible light measurements without a specialized setup. Earlier works use tomography and linear imaging formation to reconstruct density grids from visual observations (Gregson et al., 2014; Okabe et al., 2015) and then estimate velocity using physical priors (Gregson et al., 2014; Eckert et al., 2018). This line of work has been extended to joint optimization for both velocity and density (Eckert et al., 2019) to improve reconstruction quality. Since this is a highly unconstrained problem, synthesized view supervision is proposed to regularize the reconstruction (Zang et al., 2020). Franz et al. (2021) introduce a density-based rendering formation with a global differentiable simulation framework. They use manually calibrate lighting directions and fluid source location to allow consistent differentiable simulation optimization. However, these methods ignore the visual appearance, suffering from the inherent visual ambiguity of fluid density and appearance.

**Neural dynamic scene representations.** Using neural networks as implicit visual scene representations have been made effective and popular. Earlier works use neural representations for geometry (Park et al., 2019; Mescheder et al., 2019) and visual appearance (Sitzmann et al., 2019). As a seminal work, neural radiance fields (NeRFs) (Mildenhall et al., 2020) incorporate volume rendering to learn implicit geometry and appearance from images. Follow-ups of NeRFs extend applications from novel view synthesis to dynamic scene reconstruction (Li et al., 2021; Pumarola et al., 2021), relighting (Zhang et al., 2021; Yu et al., 2023), scene segmentation (Yu et al., 2022; Sajjadi et al., 2022), robot manipulation (Le Cleac'h et al., 2023; Tian et al., 2023), system identification (Li et al., 2023), etc.

Generic dynamic NeRFs incorporate deformation fields (Park et al., 2021b; Pumarola et al., 2021; Park et al., 2021a) or scene flow fields (Li et al., 2021; Du et al., 2021; Xian et al., 2021; Li et al., 2022) to represent motion. These motion representations essentially rely on stable visual features such as color gradients and edges, and thus they are not suitable for uncovering fluid flows which generally do not exhibit stable visual features. Notably, (Chu et al., 2022) is the most relevant work to ours. Chu et al. (2022) propose the physics-informed neural fields (PINF) that incorporate physics-informed losses (Raissi et al., 2019) to reconstruct fluid flows from sparse-view videos with learned priors from synthetic data. However, PINF does not consider the inherent physical structures of real fluid flows and thus only reconstructs laminar flows. Our HyFluid allows uncovering turbulent real flows by novel physics-based losses and the hybrid neural velocity representation.

**Learning fluid dynamics.** Learning fluid dynamics holds the promise to accelerate simulation and aid fluid reconstruction. Earlier work on learning fluid dynamics to accelerate simulation includes using convolutional networks to evolve fluid states (Tompson et al., 2017; Ummenhofer et al., 2019; Prantl et al., 2022) or learning latent simulation (Wiewel et al., 2019; Kim et al., 2019), supervised by 3D

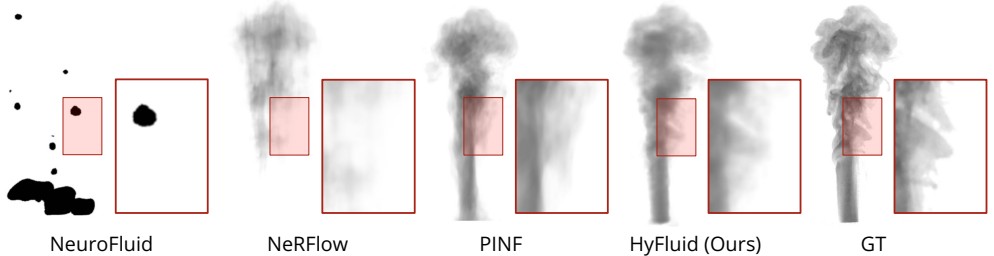

Figure 3: Visualization of novel view synthesis results on a real capture. Note that ours more faithfully recovers the density distribution of the fluid without floaters or missing regions.

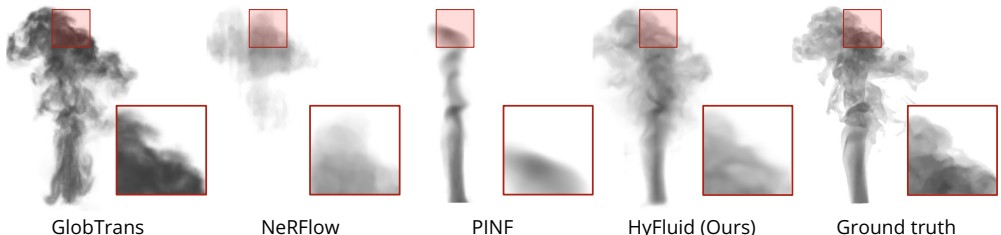

Figure 4: Visualization of re-simulation results on a real capture. Ours synthesizes reasonable re-simulation.

fluid simulation data. While these methods learn from 3D fluid data, a few recent work such as Guan et al. (2022) and Liu et al. (2023) learn simulators from multi-view videos. However, both of them assume to have groundtruth reconstructed initial fluid states, restricting the applications to synthetic data. Thus, it still remains open how existing methods may help fluid reconstruction. An exception is Deng et al. (2023) that use low-dimensional vortex particles to represent fluid velocity fields and thus allow reconstructing fluid velocity from a single video, yet their formulation is restricted to 2D domain due to complex vortex stretching in 3D vortex dynamics.

# 5 Experiments

**Datasets.** We use both real captures and synthetic simulation for evaluation. For real captures, we use the ScalarFlow dataset (Eckert et al., 2019) which consists of videos of buoyancy-driven rising smoke plumes. We use the first five scenes from the real captures. For each scene, there are five video recordings. The five cameras are fixed-position throughout capture and distributed evenly across a $120°$ arc centered at the rising smoke. Each video has 150 frames with $1062 \times 600$ resolution. These videos have been post-processed to remove backgrounds. We follow Chu et al. (2022) to use 120 frames for each video. For each scene, we use four videos for training, and one held-out video for testing (i.e., as the groundtruth for the novel view).

Synthetic simulation allows evaluating 3D velocity and density against the groundtruth. We use ScalarFlow synthetic dataset generation code (Eckert et al., 2019). We generate five examples with different inflow source with higher viscosity and another five examples with lower viscosity.

**Evaluation metrics.** For real fluid flows we do not have volumetric 3D groundtruth, so we use novel view video rendering to evaluate the reconstruction quality. In particular, we use the following three tasks: novel view video synthesis, novel view re-simulation, and novel view future prediction. Re-simulation and future prediction require high-fidelity velocity reconstruction and thus provide a means to evaluate velocity reconstruction. We use peak signal-noise ratio (PSNR), structural similarity index measure (SSIM), and the perceptual metric LPIPS (Zhang et al., 2018).

For synthetic data, we can evaluate the reconstruction results against the simulation groundtruth. Since the simulation groundtruth are up to a scale, we use scale-invariant RMSE to measure the performance. We only compute metrics where groundtruth density is greater than 0.1 to rule out empty space (which is otherwise dominant) for clearer quantitative comparison. In particular, we consider volumetric density error (by querying density networks at the simulation grid points) to

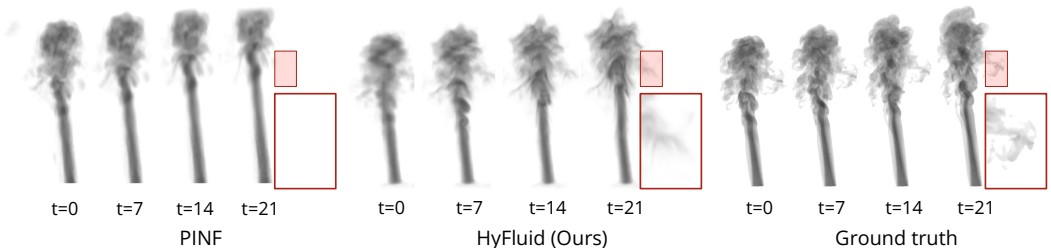

t=0  t=7  t=14  t=21
PINF

t=0  t=7  t=14  t=21
HyFluid (Ours)

t=0  t=7  t=14  t=21
Ground truth

Figure 5: Visualization of future prediction results on a real capture. Notice how our plume expands while PINF (Chu et al., 2022) only moves the plume up.

| Models | Novel view synthesis | | | Re-simulation | | | Future prediction | | |
|---|---|---|---|---|---|---|---|---|---|
| | PSNR↑ | SSIM↑ | LPIPS↓ | PSNR↑ | SSIM↑ | LPIPS↓ | PSNR↑ | SSIM↑ | LPIPS↓ |
| NeRFlow (Du et al., 2021) | 18.01 | 0.7880 | 0.1958 | 17.20 | 0.8174 | 0.1658 | - | - | – |
| GlobTran (Franz et al., 2021) | 25.97 | 0.9312 | **0.0783** | 24.55 | 0.8988 | **0.1017** | - | - | - |
| NeuroFluid (Guan et al., 2022) | 22.41 | 0.8452 | 0.1560 | - | - | - | - | - | - |
| PINF (Chu et al., 2022) | 29.55 | 0.9244 | 0.0881 | 24.97 | 0.9109 | 0.1278 | 24.19 | 0.8366 | 0.2155 |
| HyFluid (ours) | **31.14** | **0.9330** | 0.0966 | **28.37** | **0.9158** | 0.1171 | **26.12** | **0.8448** | **0.1968** |

Table 1: Comparison on renderings for three tasks on real captures.

evaluate density prediction, and warp error (i.e., using velocity to advect density and comparing to GT density) to evaluate both density and velocity prediction.

**Baselines.** We consider three recent flow reconstruction methods: NeRFlow (Du et al., 2021), a neural dynamic reconstruction method for fluid reconstruction; GlobTran (Franz et al., 2021), a grid-based fluid reconstruction method; and PINF (Chu et al., 2022), the latest neural fluid reconstruction method. We also compare to NeuroFluid (Guan et al., 2022), a recent fluid dynamics learning approach. Since NeuroFluid requires having 3D density and velocity at the first frame, we use PINF reconstructed fields for it. We leave training details and compute resources in Appendix.

## 5.1 Novel view video synthesis

In this task, we aim to evaluate the inferred density fields by synthesizing novel view videos. We use all 120 frames from the held-out test view for all scenes and report the average numbers. As shown in Table 1, our method is comparable to the state-of-the-art method PINF (Chu et al., 2022). Notice that ours is better in PSNR which is an objective metric that measures the optical similarity of the synthesized image and the groundtruth. GlobTrans (Franz et al., 2021) uses adversarial loss together with a specific rendering formulation, which may contribute to better perceptual LPIPS performances. Yet, this compromises objective fidelity as reflected by lower PSNR. Qualitative results in Fig. 3 showcase the capability of our method to accurately reconstruct density fields, which paves the way for the inference of high-fidelity physically-plausible velocity fields.

## 5.2 Re-simulation

The re-simulation task accesses the ability of the models to infer the velocity fields accurately. Specifically, we use the reconstructed density from the first frame and advect the density across time to the last frame using the learned velocity. We use the standard MacCormack method (Selle et al., 2008) for advection. As for fluid source, we use the reconstructed density field in the bottom of the scenes. Results in Table 1 and Fig. 4 show that our method generates reasonable results, whereas PINF (Chu et al., 2022) and NeRFlow (Du et al., 2021) completely fail.

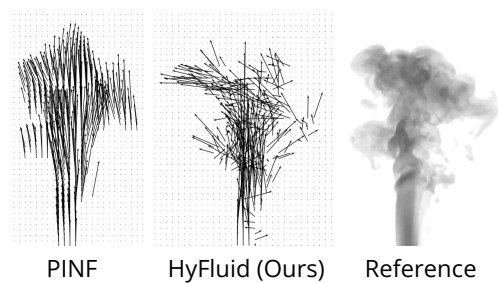

PINF    HyFluid (Ours)    Reference

Figure 6: Velocity slice on a real capture.

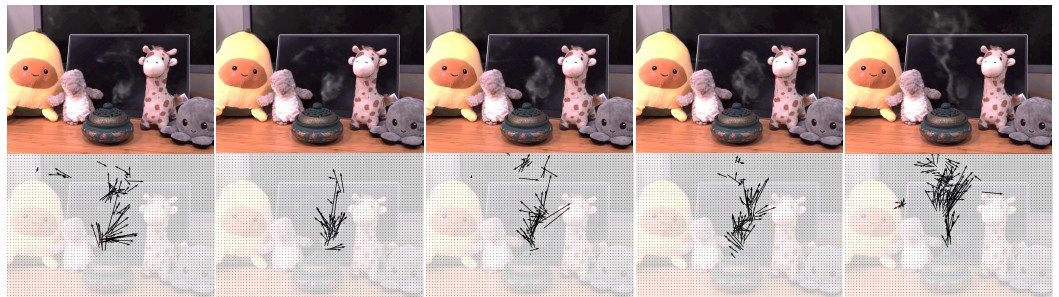

Figure 7: Visualization of velocity field inferred by HyFluid from a real 3-view capture.

| Low vis. | Den. err.↓ | Warp err.↓ | PSNR↑ | SSIM↑ | LPIPS↓ |
|---|---|---|---|---|---|
| PINF | 5.01 | 4.84 | 23.43 | 0.8555 | 0.2153 |
| Ours | **2.94** | **3.37** | **27.28** | **0.8616** | **0.1285** |
| **High vis.** | Den. err.↓ | Warp err.↓ | PSNR↑ | SSIM↑ | LPIPS↓ |
| PINF | 4.91 | 4.93 | 27.26 | **0.8728** | 0.1537 |
| Ours | **2.85** | **3.21** | **28.57** | 0.8670 | **0.1233** |

Table 2: Evaluation on synthetic data of different levels of viscosity.

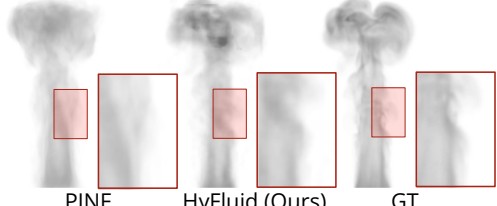

Figure 8: Novel view re-simulation on synthetic data.

We further visualize the velocity field by slicing it in Fig. 6, which demonstrates that our method effectively captures the velocity details that are missed by the baseline methods, which tend to default to trivial solutions where all the velocity is upward-directed.

### 5.3 Future prediction

In the future prediction task, we extrapolate the fluid motion into the future based on the inferred velocity fields at a single frame. In particular, we follow the standard grid-based fluid simulation (Fedkiw et al., 2001) to evolve the velocity fields by self-advecting them using the MacCormack method, followed by a projection step to ensure the divergence-free constraint. Table 1 shows that our method predicts future frames that are quantitatively better than PINF. Fig. 5 illustrates that compared to PINF, the future fluid state predicted by our method maintains its original structure and continues to follow a natural upward trajectory, and also produces natural details. These results underline the robustness of our method in inferring complex fluid velocity.

### 5.4 Inference on in-the-wild real capture

While ScalarFlow dataset provides real captures of plumes from a controlled environment, we are also interested in inferring fluid fields from uncontrolled in-the-wild videos. We use three cameras to capture sparse multi-view videos of a lit incense, and we show a visualization of the inferred density (top row) and velocity (bottom row) in Fig. 7. While the fluid in this example is much thinner than plumes, HyFluid allows plausible inference of fluid fields.

### 5.5 Evaluation on synthetic data

We include synthetic examples for evaluating 3D density and velocity fields against the groundtruth. We compare to PINF which has shown state-of-the-art results in 3D fluid fields reconstruction. We show 3D fields evaluation results together with novel view re-simulation results in Table 2 and Fig. 8. We observe that ours outperforms PINF on both 3D fields and 2D rendering, especially in preserving the vortical structures and intricate shape details. This is validated by larger performance margins in low-viscosity examples, since there is more turbulence that is hard to capture by PINF.

### 5.6 Additional applications

The inherent properties of neural representation in our method allow for its applicability in a variety of additional contexts, such as editing and scene composition. As seen in Fig. 1, the results obtained

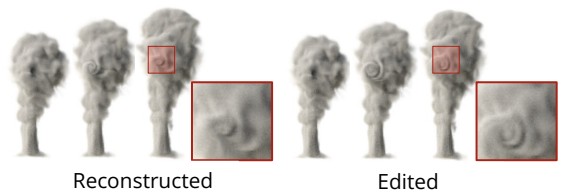

Reconstructed          Edited

Figure 9: Visualization of our turbulence editing by intensifying the learned vortex particles on a real capture.

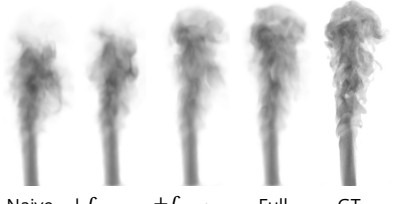

Naive    $+\mathcal{L}_{\text{laminar}}$  $+\mathcal{L}_{\text{project}}$  Full    GT

Figure 10: Ablation study by re-simulation.

from our method can be effortlessly integrated into dynamic NeRF scenes (Sara Fridovich-Keil and Giacomo Meanti et al., 2023) or imported into professional-grade graphics software.

Our hybrid neural velocity representation naturally supports turbulence editing in re-simulation. In particular, we multiply our vortex particle intensity by a factor of 4 and showcase a visualization of edited re-simulation in Fig. 9. This shows that the high-quality velocity estimation from our hybrid neural velocity representation allows easily synthesizing vortical flow details.

### 5.7 Ablation study

We show qualitative results in Fig. 10 to evaluate the proposed physics-based losses, the vortex particle-driven velocity representation, and the joint training supervision. Starting from "Naive" where we only use the rendering loss for recovering density and the density loss for recovering velocity, we gradually add back laminar loss ("$+\mathcal{L}_{\text{laminar}}$"), projection loss ("$+\mathcal{L}_{\text{project}}$"), and vortex particle-driven residual velocity ("Full"). From Fig. 10 we observe that our physical losses allow better velocity recovery. In particular, the projection loss enables recovering physically correct velocity that reconstructs the plume shape in re-simulation. Vortex particle-driven velocity allows capturing more details and thus more consistent re-simulation.

## 6 Conclusion

In this work, we study recovering fluid density and velocity from sparse multi-view videos. We propose hybrid neural fluid fields (HyFluid) which features physics-based losses to address inherent visual ambiguities and a hybrid neural velocity representation for capturing the complex turbulent fluid flow. We show that our simple designs can already lead to physically plausible estimation of fluid fields that supports applications such as re-simulation, editing, and dynamic scene composition.

**Limitations.** Our method assumes inviscid fluid and only considers gas but not liquid. Liquid features free surface that may require further physical modeling and constraints.

**Acknowledgments.** This work was in part supported by the Toyota Research Institute (TRI), NSF RI #2211258, ONR MURI N00014-22-1-2740, the Stanford Institute for Human-Centered AI (HAI), and Google. B. Zhu acknowledges NSF IIS #2313075, IIS #2106733, and CAREER #2144806.

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

# A  Appendix: Implementation details

**Model architecture.** For both the density field $\sigma(x, y, z, t)$ and the base velocity field $\mathbf{u}_{\text{base}}(x, y, z, t)$ we use a 4D extension of iNGP (Müller et al., 2022). Specifically, we add a temporal dimension to the original static iNGP. For the spatial dimensions, we use a base resolution 16 and the finest resolution 256. For the temporal dimension we set the finest resolution to 128 which is comparable to the number of video frames (120 frames), as higher resolutions create hash encoding features that are never directly supervised and can lead to unstable gradients.

**Training and hyperparameters.** Our training can be divided into three stages. In the first stage, we pre-train our density network $\sigma(x, y, z, t)$ and the constant learnable appearance (radiance) $L_e$ using the rendering loss only, which forms an initial estimate of both quantities. In the second stage, we jointly train the density $\sigma(x, y, z, t)$, appearance $L_e$, and the base velocity $\mathbf{u}_{\text{base}}(x, y, z, t)$ using the full loss. Finally, we jointly train the density $\sigma(x, y, z, t)$, appearance $L_e$, and the vortex particles $\{I_p\}$. We use an Adam optimizer with a learning rate 0.01. In the first stage, we train the density and radiance for $200,000$ iterations. In the second stage, we jointly train the model for $50,000$ iterations. In the third stage, we do training for $5,000$ iterations. We empirically set the loss weights to $\beta_{\text{render}} = 10,000$, $\beta_{\text{density}} = 0.001$, $\beta_{\text{proj}} = 1$, $\beta_{\text{laminar}} = 10$. For the laminar loss, we set the coefficient $\gamma = 0.2$.

**Projection loss.** For the projection routine, we use a multi-grid preconditioned conjugate gradient (MGPCG) solver (Ashby & Falgout, 1996). We implement it using Taichi (Hu et al., 2019) language. We use three levels of multi-grid, with the spatial resolution $128^3$. Note that we use the same resolution for our re-simulation and future prediction experiments for all compared methods.

**Seeding strategy.** In our experiments, we seed 50 vortex particles for each scene. To place initial vortex particles, we take the insight from vortex confinement methods (Fedkiw et al., 2001). We densely sample continuous spatio-temporal locations and compute curl values from the trained base velocity network, and then we place vortex particles at locations with the highest curl values.

**Compute usage.** We train our model on a single A100 GPU for around 9 hours in total. The first stage takes half an hour, the second stage takes around 8 hours (this is significantly slower than the first stage as it requires computing derivatives for the density loss), and the third stage takes half an hour.

