# OpenReview forum: "Inferring Hybrid Neural Fluid Fields from Videos"
_NeurIPS.cc/2023/Conference — NeurIPS 2023 poster_

### Official Review · Reviewer_MRuw · 2023-07-03

**Soundness:** 3 good
**Presentation:** 3 good
**Contribution:** 3 good
**Rating:** 6
**Confidence:** 5

**Summary:**

The paper proposes a novel approach for recovering the density and velocity fields of inviscid fluids from sparse multiview videos. The model has two main contributions:
- First, it incorporates physics-based losses to enforce the inference of a physically plausible velocity field that is divergence-free and drives the transport of density. This helps to deal with the visual ambiguities of fluid velocity.
- Second, the model provides a hybrid neural velocity representation, which consists of a base neural velocity field capturing most irrotational energy and a vortex particle-based velocity modeling residual turbulent velocity. This representation enables the recovery of vortical flow details.

**Strengths:**

Originality: This paper primarily tackles the visual ambiguities in inverse rendering techniques for fluid data, focusing on resolving visual ambiguities. The integration of physics-based losses into the volume rendering framework is a rational approach. The results in Figure 8 illustrate the effectiveness of the newly proposed learning constraints in significantly improving the accuracy of the reconstruction. Moreover, the introduction of a hybrid representation of velocity fields and particle-based vortex flow showcases originality in the methodology.

Significance: The proposed model makes a significant contribution to the visual understanding of fluids, particularly smoke, fog, and gas.


**Weaknesses:**

For methodology:

1. One contribution of this paper is to incorporate new forms of physical constraints in the framework of inverse rendering. However, the general idea is not entirely novel as previous work by Chu et al. (2022) has also presented a similar (albeit different) approach, which somewhat weakens the technical novelty of this paper. For example, using the density transport equation from incompressible fluid or NS equation as a loss function has been employed in other related papers. Please refer to the work from Baieri et al. (2023) and Li et al. (2023).

- [Baieri et al., 2023] Fluid Dynamics Network: Topology-Agnostic 4D Reconstruction via Fluid Dynamics Priors. Arxiv, 2023.
- [Li et al., 2023] PAC-NeRF: Physics Augmented Continuum Neural Radiance Fields for Geometry-Agnostic System Identification. ICLR, 2023.

2. The constraint imposed on emitting radiance in HyFluid does present limitations in its applicability to real-world scenarios. The requirement of a constant emitting radiance may hinder accurate recovery in situations where spatially-varying lighting exists in the scene.

Lacking references:

3. Some closely related work seems to be missing, such as NeuroFluid (Guan et al., 2022) and PAC-NeRF (Li et al., 2023), both of which also focus on visual physical inference through inverse rendering. It is important to acknowledge these works as they contribute to the existing body of literature in this field and provide valuable insights and techniques for comparison and benchmarking purposes.

- [Guan et al., 2022] NeuroFluid: Fluid Dynamics Grounding with Particle-Driven Neural Radiance Fields. ICML, 2022.

For experiments:

4. Is the proposed method limited to handling only inviscid fluids? It would be beneficial to evaluate the proposed method in a broader range of fluid scenarios, including different types of flows (e.g., laminar, turbulent) and varying fluid properties (e.g., viscosity, density). This would demonstrate the generalization ability of HyFluid under diverse conditions.

5. Considering that the constraint of constant radiance may not be practical in complex real-world scenes, I highly recommend that the authors compare the reconstruction results obtained with and without the constraint.

6. The proposed model is only compared with two existing models, which is not sufficient. To make the results more convincing, the authors could incorporate more advanced neural rendering techniques designed specifically for dynamic scenes. Additionally, given that the dataset comprises synthetic fluid simulation data, it would be beneficial for the authors to provide quantitative results regarding the reconstruction of the velocity field in comparison to the ground truth on these simulated scenes.

**Questions:**

1. In the paper, a grid-based representation is used for density and velocity. When implementing $L_{density}$ and $L_{project}$, how exactly are these calculations performed? Are the loss functions computed for all grid positions, or is interpolation used to compute the losses at sampled points in space?

2. What is the difference in performance between decomposing the velocity field into a base neural velocity field and a vortex particle-based velocity, versus solely using high-frequency position embedding?

3. In the case of using physics-based loss functions in HyFluid, how does the predicted velocity field of HyFluid compare to the ground truth (GT) velocity field quantitatively? Can HyFluid be compared to other methods in terms of velocity field quantitatively?

4. The paper does not provide experimental results and discussions regarding the constraint of constant radiance. What is the capability of HyFluid with real-world scenes that exhibit spatially-varying radiance?

5. Can you provide more quantitative/ qualitative results of other baselines?

**Limitations:**

Yes

---

> ### Author Rebuttal · Authors · 2023-08-10
>
> Thank you for your insightful comments and constructive suggestions! Please see our response below.
>
> For methodology:
> 1. **Technical novelty**:
> [Chu et.al.] is indeed most relevant. Our approach incorporates novel losses including projection loss and laminar loss as well as a hybrid representation to capture turbulent velocity fields. Our technical novelty leads to better velocity reconstruction compared to [Chu et.al.] as can be seen in novel view re-simulation and velocity visualization (especially the videos in supplementary material). [Baieri et.al.] and [Li et.al.] focus on general dynamic object and they do not consider complex fluid dynamics such as turbulence. We will add these references and discussion to our paper.
>
> 2. **Spatially-varying appearance modeling**:
> Following your suggestion, we allow learning spatially and temporally varying appearances. Please see the results and discussion in the "Spatiotemporally varying appearance" in the global response above.
>
> For reference:
>
> 3. **References**:
> Thank you for the note. We will add these references to our paper. We will also add a clarification that our setting is different from NeuroFluid. NeuroFluid focuses on learning fluid dynamics from a large amount of data. Therefore, they train and evaluate the model on synthetic data. Moreover, they assume known initial state and no inflow source. In contrast, our goal is to reconstruct plausible fluid fields from real sparse multiview videos without assuming any training data.
>
> For experiments:
>
> 4. **Different viscosity level**:
> Following your suggestion, we include additional synthetic data experiments which have different viscosity levels. Please refer to the "Evaluation on different viscosity levels" in the global response above for results and discussion.
>
> 5. **Spatially-varying appearance modeling**: Please see response to 2. above.
>
> 6. **Comparison to other existing methods**:
> We include NeuroFluid and GlobTrans [Franz2021] as additional compared methods. Please refer to the "Comparison to GlobTrans" and "Comparison to NeuroFluid" in the global response above.
>
> 7. **Quantitative results regarding velocity field reconstruction**:
> We clarify that our goal is to reconstruct plausible fluid fields from real videos, and thus in our experiments in the main paper **we only use real data** (as specified in L201) which does not provide groundtruth 3D fields but only multi-view videos. To evaluate 3D fields, we include experiments on synthetic examples in the "Evaluation on synthetic data" in the global response.
>
> For questions:
>
> 1. **How to compute losses**:
> For L_density, in each training step, we randomly select one timestamp (i.e., one frame) and we sample continuous points in 3D space. For these points, we compute the first-order derivatives by auto-gradient computation provided by PyTorch. For L_project, we do not use sampling as our MGPCG solver requires a regular grid. Thus, we use regular grid of 128^3 to solve for projection. These projected velocity vectors at the regular grid points are then used to supervise the velocity network outputs at those exact regular grid points.
>
> 2. **Differences compared to solely using high-frequency position embedding**:
> Please note that the state-of-the-art neural fluid reconstruction method PINF [Chu2022] uses an advanced high-frequency position embedding [Sitzmann2020] for velocity field. Our comparison to PINF shows that our approach reconstructs better fluid fields with richer vortical details (in particular, please see the re-simulation video in supplementary material).
>
> 3. **Comparison to GT velocity**:
> Please see our response in 7. above.
>
> 4. **HyFluid on real scenes**:
> As clarified above, all our experiments use real data which inevitably has spatially-varying lighting due to global illumination. Please also see our response in 2. above for modeling spatially-varying appearance.
>
> 5. **More results of other baselines**:
> Please refer to our response in 6. above.
>
> **Reference**:
>
> [Chu2022] Chu, M., Liu, L., Zheng, Q., Franz, E., Seidel, H. P., Theobalt, C., & Zayer, R. (2022). Physics informed neural fields for smoke reconstruction with sparse data. ACM Transactions on Graphics (TOG), 41(4), 1-14.
>
> [Sitzmann2020] Sitzmann, V., Martel, J., Bergman, A., Lindell, D., & Wetzstein, G. (2020). Implicit neural representations with periodic activation functions. Advances in neural information processing systems, 33, 7462-7473.

---

> > ### Comment · Reviewer_MRuw · 2023-08-18
> > **Thanks for the reply**
> >
> > Thank you for the replies to my questions and comments. After reading the other reviews and answers, most of my concerns are addressed. I'll raise my score on this paper and look forward to its future updates with improved content.

---

> > > ### Author Response · Authors · 2023-08-18
> > > **Thank you for your updated review!**
> > >
> > > Dear Reviewer MRuw,
> > >
> > > Thank you for kindly updating your review! We will incorporate all the contents in the rebuttal to our revised manuscript.
> > >
> > > Sincerely,
> > >
> > > Authors of submission 1335

---

> ### Author Response · Authors · 2023-08-14
> **Happy to answer any further questions**
>
> Dear Reviewer MRuw,
>
> Thank you for reviewing our submission. We have posted our response per your suggestions and questions. We are happy to discuss with you and answer any further questions. We are looking forward to your feedback!
>
> Best,
>
> Authors of submission 1335

---

> ### Author Response · Authors · 2023-08-17
> **Looking forward to discussion with you**
>
> Dear Reviewer MRuw,
>
> We have posted our clarification and response. We wish our response can address your concerns and we would like to hear your updated feedback and evaluation. We are more than happy to discuss with you and answer any further questions!
>
> Best,
>
> Authors of submission 1335

---

### Official Review · Reviewer_4cV8 · 2023-07-03

**Soundness:** 2 fair
**Presentation:** 3 good
**Contribution:** 2 fair
**Rating:** 7
**Confidence:** 4

**Summary:**

The paper presents a neural reconstruction method for individual fluid flows which is re-trained for each new scene. It combines the established iNPGs for representing densities and velocities with vorticity transporting partices in a hybrid flow representation called HyFluid. A set of appearance- and physics-based losses is used to train the iNPGs and optimize vorticity and lighting in 3 steps. With this approach it is possible to recover 3D density and velocity from sparse views and enable novel-view rendering, re-simulation, and predictions of states over time.


**Strengths:**

I see the following strong points in this submission:
- The method is build on physical priors to recover a physically meaningful velocity field.
- The loss to the projected velocity to ensure divergence freeness is novel as far as I can tell.
- Using particles to transport voriticity seems novel, at least in that context.
- Provides better results then SOTA neural methods regarding re-simulation and prediction.
- The results look good, the motion of the density is coherent and also suited for re-simulation.
- The reconstruction works with simple, sparse images.


**Weaknesses:**

The treatment for vorticity seems novel, but from the provided ablations I cannot see a substantial impact:
- In Fig. 8 there seems to be a slight improvement.
- In the videos, the ablation study results of "w/o vort." and "full" look very similar.
- The difference could be more visible in the flow field, and but this is not demonstrated.
- In the videos, the novel view synthesis results of PINF [1] and HyFlow are almost identical.


**Questions:**

- Since the approach is trained on ScalarFlow [2] it would be interesting to see how the method compares to the ScalarFlow reconstructions. ScalarFlow is not using neural networks, and as such a good not-learned baseline as a single scene reconstruction method using physical priors (advection, pressure projection). Re-simulation and future prediction might also be possible there since ScalarFlow is using a fluid solver (MantaFlow) internally. Can the authors explain whether they haven't compared to the ScalarFlow 3D data despite using the 2D input images?
- It would also be interesting to compare to GlobTrans [3] as this method it is trained/optimized as a re-simulation method. Have the authors looked into this?
- The paper uses use an unseen GT view to evaluate the metrics. Have the authors tried/considered reconstructing a synthetic case to be able to make a 3D comparison? This might also show the efficacy of the flow reconstruction more clearly, since the visual appearance could be matched perfectly.
- The velocity flickers in the supplemental video, there is no smooth evolution. Is there any coupling of the velocity over time (like self-advection)?
- In the videos the velocity fields are rendered to 2D. Can the authors explain how is that done? Wouldn't a slice provide more insight in the actual motions? Averaging might blur the small-scale vorticities.

Overall, I think the paper targets an interesting direction and provides an interesting approach for a tough problem. I still somewhat on the edge regarding this paper, but I'm open to readjusting/raising my score after the rebuttal.

References:
[1] Mengyu Chu, Lingjie Liu, Quan Zheng, Erik Franz, Hans-Peter Seidel, Christian Theobalt, and Rhaleb Zayer. 318 Physics informed neural fields for smoke reconstruction with sparse data. ACM Transactions on Graphics, 2022.
[2] Marie-Lena Eckert, Kiwon Um, and Nils Thuerey. Scalarflow: a large-scale volumetric data set of real-world 330 scalar transport flows for computer animation and machine learning. ACM Transactions on Graphics (TOG), 2019.
[3] Erik Franz, Barbara Solenthaler, and Nils Thuerey. Global transport for fluid reconstruction with learned 337 self-supervision. In Proceedings of the IEEE/CVF Conference on Computer Vision and Pattern Recognition, 2021.


**Limitations:**

Limitations are briefly discussed, I don't see anything significant missing here.

---

> ### Author Rebuttal · Authors · 2023-08-10
>
> Thank you for your insightful comments and constructive suggestions! Please see our response below.
>
> - **Comparison to ScalarFlow and GlobTrans**:
> Thank you for your suggestion! Since both ScalarFlow and GlobTrans are optimization-based methods, we compare to GlobTrans which is a later work and it has shown better results than ScalarFlow. Please refer to the "Comparison to GlobTrans" in the global response above for results and discussion.
>
> - **Synthetic data for 3D evaluation**:
> Following your suggestion, we additionally include synthetic examples using ScalarFlow synthetic dataset generation code. Please refer to the "Evaluation on synthetic data" and "Evaluation on different viscosity levels" in the global response above for results and discussion.
>
> - **Self-advection of velocity**:
> We do not include a physical loss for self-advection of velocity like $\mathcal{L}=D\mathbb{u}/Dt-\mathbb{f}$ (where $\mathbb{f}$ denotes external force) as we empirically found that it often leads to oversmoothed velocity fields. This may be due to that the material derivative term for velocity $D\mathbb{u}/Dt=\partial \mathbb{u}/\partial t+\mathbb{u}\cdot\nabla \mathbb{u}$ admits local trivial solutions where the velocity is spatiotemporally constant. In optimization-based methods such as GlobTrans, this local trivial solution is solved by a global optimization. However, in neural continuous reconstruction, this is not straightforward to address and we leave it as future exploration. Actually, we suspect that this is the reason that PINF (which uses a velocity advection loss) reconstructs only laminar flows for real videos.
>
> - **Render velocity field as slice**:
> We rendered velocity field to 2D by projecting every 3D velocity vector to the camera plane and then use volumetric rendering to integrate them. This indeed smoothes the visualization. Following your suggestion, we additionally include slice rendering in Figure R6 in the global response PDF. From the comparison we can see that our velocity field recovers more vortical details than PINF. We will include this figure in our revised paper.

---

> > ### Comment · Reviewer_4cV8 · 2023-08-17
> > **Post rebuttal**
> >
> > I thank the authors for the clarifications, good to see the synthetic results. Unlike mentioned in other reviews, this now shows both real and synthetic results, so I’d be happy to support acceptance, and I’ll raise my score.

---

> > > ### Author Response · Authors · 2023-08-17
> > > **Thank you for your updated review!**
> > >
> > > Dear Reviewer 4cV8,
> > >
> > > Thank you for kindly updating your review! We will incorporate all the contents in the rebuttal to our revised manuscript.
> > >
> > > Sincerely,
> > >
> > > Authors of submission 1335

---

> ### Author Response · Authors · 2023-08-14
> **Happy to answer any further questions**
>
> Dear Reviewer 4cV8,
>
> Thank you for reviewing our submission. We have posted our response per your suggestions and questions. We are happy to discuss with you and answer any further questions. We are looking forward to your feedback!
>
> Best,
>
> Authors of submission 1335

---

### Official Review · Reviewer_3dGa · 2023-07-06

**Soundness:** 2 fair
**Presentation:** 2 fair
**Contribution:** 2 fair
**Rating:** 5
**Confidence:** 5

**Summary:**

This paper presents an innovative neural dynamic reconstruction method that achieves good results in recovering fluid density and velocity fields through the introduction of physical constraints and a hybrid neural velocity representation. Despite some weaknesses and issues, this method has significant implications for the in-depth study and resolution of fluid dynamics problems. Future work could focus on refining the method and validating and applying it to a wider range of fluid scenarios.

**Strengths:**

This method proposes a new approach to neural dynamic reconstruction that can simultaneously recover fluid density and velocity fields, overcoming the challenges posed by the visual ambiguities of fluid velocity in existing methods.
Physics-based losses are introduced to enforce a divergence-free velocity field, driving the transport of density and enhancing the accuracy of velocity estimation.
A hybrid neural velocity representation is designed, incorporating a base neural velocity field that captures most irrotational energy and a vortex particle-based velocity that models residual turbulent velocity. This enables the method to effectively recover vortical flow details.

**Weaknesses:**

The paper lacks detailed presentation of experimental results and quantitative evaluations, thus lacking sufficient validation of the method's performance.
There is no comparison with other methods in addressing the visual ambiguity of fluid velocity, making it difficult to assess the method's advantages and disadvantages comprehensively.

**Questions:**

Further investigation is needed to determine whether the method can handle complex fluid scenarios such as non-Newtonian or multiphase fluids in practical applications.

**Limitations:**

The paper does not mention the computational resource requirements, such as computation time and memory consumption, which are important factors for the feasibility of practical applications.

---

> ### Author Rebuttal · Authors · 2023-08-10
>
> Thank you for your time and comments. Please see our response below.
>
> - **Detailed presentation of experiment results**:
> We clarify that we aim at reconstructing plausible fluid velocity field from real videos to allow re-simulation and future prediction. For real videos, it is very challenging to collect groundtruth 3D density and velocity.
> Therefore we evaluate on the applications including novel view video synthesis, novel view re-simulation, and novel view future prediction. For each of them, we show detailed quantitative results in Table 1 (three metrics for each task) and qualitative results in Figure 3, 4, 7 in our main paper. In addition, we show qualitative results on turbulence editing and velocity recovery in Figure 5 and Figure 6. These downstream applications reflect that our approach can reconstruct plausible real fluid fields.
>
> - **Additional quantitative evaluation**:
> In addition to the experiments on real fluid videos, we include new experiments on synthetic fluid scenes. These scenes provide 3D groundtruth density and velocity, allowing quantitative evaluations on them. Please refer to the "Evaluation on synthetic data" and "Evaluation on different viscosity levels" in the global response for results and discussion.
>
> - **No comparison with other methods in addressing the visual ambiguity of fluid velocity**:
> We respectively disagree with this. We clarify that we have comparisons to PINF [Chu2022] and NeRFlow [Du2021], both of which aim to address visual ambiguity of velocity/flow estimation from real videos, and they both showcase plausible reconstruction on fluid scenes in their results. In particular, PINF [Chu2022] is the state-of-the-art fluid reconstruction method which tries to address visual ambiguity by physics-informed losses similar to Physics-informed Neural Networks (PINN) [Raissi2019]. PINF shows extensive results in synthetic scenes and a few real examples. NeRFlow approaches this by a set of temporal consistency losses and showcase fluid reconstruction in their "milk pouring" scene. Our comparison to them in Table 1 and Figure 3, 4, 5, 7 clearly demonstrate that our approach achieves better results than these existing methods.
>
> - **Additional comparison with other methods**:
> In addition to both PINF and NeRFlow, we compare to NeuroFluid [Guan2022], and GlobTrans [Franz2021]. NeuroFluid learns fluid dynamics to address velocity ambiguity. GlobTrans aims for reconstructing fluid fields using advection constraints and regularization terms to solve visual ambiguity of fluid velocity. Please refer to the "Comparison to NeuroFluid" and "Comparison to GlobTrans" in the global response for results and discussion.
>
> - **Computation time and memory consumption**:
> As we noted in L215 in our main paper, we leave computation resource usage in our supplementary material. As in L29 in our supplementary material, we train our model on a single A100 GPU (the GPU memory usage is around 30GB) for around 9 hours in total.
>
> **Reference**:
>
> [Chu2022] Chu, M., Liu, L., Zheng, Q., Franz, E., Seidel, H. P., Theobalt, C., & Zayer, R. (2022). Physics informed neural fields for smoke reconstruction with sparse data. ACM Transactions on Graphics (TOG), 41(4), 1-14.
>
> [Du2021] Du, Y., Zhang, Y., Yu, H. X., Tenenbaum, J. B., & Wu, J. (2021, October). Neural radiance flow for 4d view synthesis and video processing. In 2021 IEEE/CVF International Conference on Computer Vision (ICCV) (pp. 14304-14314). IEEE Computer Society.
>
> [Raissi2019] Raissi, M., Perdikaris, P., & Karniadakis, G. E. (2019). Physics-informed neural networks: A deep learning framework for solving forward and inverse problems involving nonlinear partial differential equations. Journal of Computational physics, 378, 686-707.
>
> [Guan2022] Guan, S., Deng, H., Wang, Y., & Yang, X. (2022, June). Neurofluid: Fluid dynamics grounding with particle-driven neural radiance fields. In International Conference on Machine Learning (pp. 7919-7929). PMLR.
>
> [Franz2021] Franz, E., Solenthaler, B., & Thuerey, N. (2021). Global transport for fluid reconstruction with learned self-supervision. In Proceedings of the IEEE/CVF Conference on Computer Vision and Pattern Recognition (pp. 1632-1642).

---

> ### Author Response · Authors · 2023-08-14
> **Happy to answer any further questions**
>
> Dear Reviewer 3dGa,
>
> Thank you for reviewing our submission. We have posted our clarification and response to your suggestions and questions. We are happy to discuss with you and answer any further questions. We are looking forward to your feedback!
>
> Best,
>
> Authors of submission 1335

---

> ### Author Response · Authors · 2023-08-17
> **Looking forward to discussion with you**
>
> Dear Reviewer 3dGa,
>
> We have posted our clarification and response. We wish our response can address your concerns and we would like to hear your updated feedback and evaluation. We are more than happy to discuss with you and answer any further questions!
>
> Best,
>
> Authors of submission 1335

---

> > ### Comment · Reviewer_3dGa · 2023-08-19
> >
> > I acknowledge that the author has addressed my concerns well, and I recommend a weak acceptance of this paper.

---

> > > ### Author Response · Authors · 2023-08-21
> > > **Thank you for updating your review!**
> > >
> > > Dear Reviewer 3dGa,
> > >
> > > Thank you for updating your review! We will incorporate all the contents in the rebuttal to our revised manuscript.
> > >
> > > Sincerely,
> > >
> > > Authors of submission 1335

---

### Official Review · Reviewer_qUcR · 2023-07-08

**Soundness:** 3 good
**Presentation:** 2 fair
**Contribution:** 3 good
**Rating:** 5
**Confidence:** 5

**Summary:**

This paper works on reconstructing fluid density and velocity from multi-view videos. The main idea is to inject visual clues with NeRF. They propose some physics-based regularization terms to deal with visual ambiguity that video can not reflect the inner fluid states.

**Strengths:**

Good motivation: Visual ambiguity indeed is the critical problem that needs to be solved in this setting.

Novel framework: Introducing physical losses is a not-easy but natural point. The introduced losses are general to inspire other related researchers.




**Weaknesses:**


1. Setting is not new. Indeed, reconstructing fluid velocity and density from videos has been studied in NeuroFluid (ICML 2022). I didn't find the introduction or comparison. Why?

2. The proposed method is not convincing to me.

   -  The proposed physics-based losses can only play a regularization role in the training. To solve the visual ambiguity, the framework must introduce direct supervision of the fluid velocity. Nevertheless, ambiguity still exists.

   - And, the key question is how to get the initial state of the fluid, which decides the performance of this work. The authors **should** clarity this point.

3. Experiments are incomplete, lacking key results.

    - Experiments just show the render results and velocity results. How about density? The authors claim that they can recover fluid density but not shown in the experiments.

    - How did you evaluate the render results? Directly render fluid from the view angle used in the training stage? Please claim them before your experiments.
    - Only evaluate the stoke with a similar environment. Can you show more examples, with more complex shapes and materials?


**Questions:**

For details, please refer to the weaknesses section.

**Limitations:**

yes

---

> ### Author Rebuttal · Authors · 2023-08-10
>
> Thank you for your time and comments! Please see our response below.
>
> - **Relation to NeuroFluid**:
> We clarify that our setting is different from NeuroFluid. NeuroFluid focuses on learning fluid dynamics from a large amount of data. Therefore, they train and evaluate the model on synthetic data, yet the generated data distribution can be very different from real data to generalize. Moreover, they assume known initial state and no inflow source, which do not hold in some real scenes such as smoke plume scenes we used. In contrast, our goal is to reconstruct plausible fluid fields from real sparse multiview videos without assuming additional training data, so as to facilitate applications on real fluid videos such as novel view synthesis, re-simulation, future prediction, and turbulence editing.
>
> - **Comparison to NeuroFluid**:
> We show a comparison to NeuroFluid in the "Comparison to NeuroFluid" in the global response above. Please note that since we use a real dataset for evaluation (as specified in L201 in our main paper), we do not have groundtruth initial states that NeuroFluid requires as input. Therefore, we use the previous state-of-the-art method PINF [Chu2022] to reconstruct the first frame for the initial state of NeuroFluid. We will add reference and discussion to our paper.
>
> - **Physics-based losses for visual ambiguity and supervision from velocity**:
> We clarify that we aim at reconstructing plausible fluid velocity field from real videos to allow re-simulation and future prediction. We do not assume training data of groundtruth velocity as it is very scarce in real scenes. Therefore, we propose physics-based losses to regularize the recovery of fluid velocity such that they are physically plausible for the downstream applications.
>
> - **Initial state of fluid**:
> Different from NeuroFluid which requires initial states and learns fluid dynamics, we aim at reconstructing the fluid fields. Thus, the "initial state" is our model output rather than input.
>
> - **Evaluating 3D density**:
> We clarify that all our experiments are done on **real videos** which do not have groundtruth 3D density and velocity fields. Therefore, we indirectly evaluate them by downstream applications including novel view synthesis, re-simulation, and future prediction. Following your suggestion, we additionally evaluate our method on synthetic examples which provide groundtruth 3D density. Please refer to the "Evaluation on synthetic data" in the global response above to see the results and discussion.
>
> - **Evaluating rendering results**:
> We clarify that we evaluate rendering results in a hold-out novel view that is unseen during training (L205-L206 in the main paper). In particular, each example in the ScalarFlow real dataset and synthetic dataset has 5 views. We take 4 views for training, and 1 view for testing. We use this training-testing split for all our experiments including novel view video synthesis, novel view re-simulation, and novel view future prediction.
>
> - **Experiments on different examples**:
> Please note that the additional synthetic data in "Evaluation on different viscosity levels" in the global response demonstrate different material properties (e.g., they are more viscid than real smokes), shapes and inflows. We believe these additional examples provide more diverse evaluations.
>
> **Reference**:
>
> [Chu2022] Chu, M., Liu, L., Zheng, Q., Franz, E., Seidel, H. P., Theobalt, C., & Zayer, R. (2022). Physics informed neural fields for smoke reconstruction with sparse data. ACM Transactions on Graphics (TOG), 41(4), 1-14.

---

> ### Author Response · Authors · 2023-08-14
> **Happy to answer any further questions**
>
> Dear Reviewer qUcR,
>
> Thank you for reviewing our submission. We have posted our response per your suggestions and questions. We are happy to discuss with you and answer any further questions. We are looking forward to your feedback!
>
> Best,
>
> Authors of submission 1335

---

> > ### Comment · Reviewer_qUcR · 2023-08-14
> > **Thanks for your response!**
> >
> > Dear author,
> >
> > Thanks for your efforts, which answer most of my questions. But, there are two questions I need to discuss further.
> > 1. I think only testing on the ScalarFlow dataset is limited. Could you compare with PINF on other real scenes, such as predicting water? Otherwise, you can explain why evaluating on the ScalarFlow is sufficient.
> > Minor: could you please show some image samples of the scene you used?
> > 2. Could you please describe the angle interval of the 5 views in the novel-view synthesis?
> >
> > Thanks!

---

> > > ### Author Response · Authors · 2023-08-14
> > > **Thank you for your questions!**
> > >
> > > Dear Reviewer qUcR,
> > >
> > > Thank you for your further questions. Please find our responses below. We are happy to discuss further if you have any additional questions!
> > >
> > > **1. Predicting real water scenes**:
> > >
> > > We note that free-surface fluid like water has more complex dynamics and different appearance properties, making it a different problem than gases. In particular:
> > >
> > > **Complex dynamics (simulation)**: Free-surface fluid dynamics needs to consider interface dynamics, i.e., how the free surface interacts with its surroundings such as container and air. This also includes complex dynamic phenomena like wave breaking, droplet formation and capillary wave. Moreover, the NS equation holds only within the interior domain, which changes its shape along the fluid evolution.
> > >
> > > **Reflective appearance (rendering)**: Water has different appearance properties, such as strong reflection, refraction, and high transparency. This makes it a difficult to be modeled using direct volumetric rendering (We suspect this is why NeuroFluid which also uses volumetric rendering only tests on synthetic semi-opaque water without background or containers).
> > >
> > > Given these differences in both simulation and rendering, we consider free-surface fluid out of our scope. We follow existing fluid reconstruction works, which mostly focus on gases. We follow them to test on synthetic scenes and the ScalarFlow real dataset [Eckert2019, Zang2020, Franz2021, Chu2022], as ScalarFlow features a tractable representative real setting. Please also note that we have added experiments on synthetic data which demonstrates different material properties.
> > >
> > > **2. Image samples**:
> > >
> > > We have uploaded an image example of a ScalarFlow real scene in this anonymous link: https://ibb.co/phFwbNt Please note that this has been post-processed to remove background as done in existing work [Eckert2019, Franz2021, Chu2022].
> > >
> > > **3. Angle intervals**:
> > >
> > > Please kindly refer to the Figure 1 in this website: https://ge.in.tum.de/publications/2019-scalarflow-eckert/ for the capture setup. The five cameras are placed evenly in a 120 degree arc. We use the middle camera as test view and the other four as training views.
> > >
> > > **Reference**:
> > >
> > > [Eckert2019] ScalarFlow: a large-scale volumetric data set of real-world scalar transport flows for computer animation and machine learning. TOG2019
> > >
> > > [Zang2020] Tomofluid: Reconstructing dynamic fluid from sparse view videos. CVPR2020
> > >
> > > [Franz2021] Global transport for fluid reconstruction with learned self-supervision. CVPR2021
> > >
> > > [Chu2022] Physics informed neural fields for smoke reconstruction with sparse data. TOG2022

---

> > > > ### Comment · Reviewer_qUcR · 2023-08-17
> > > > **Thanks for your kind response!**
> > > >
> > > > Dear authors,
> > > >
> > > > Sorry for the late reply! Your response solved all my questions. I reconsider the paper and list my justification here.
> > > >
> > > > Novelty:
> > > > [+] Although inversing physics from a video is not a new setting, the authors proposed some interesting ideas on free-surface fluid.
> > > > [-] The title didn't clearly express your contribution. Actually, the fluid is the free-surface fluid.
> > > >
> > > > Experiments:
> > > > [+] The evaluation setting is not precise, but the authors explained them in the rebuttal. I think the response is clear. I encourage authors to include them in the next version.
> > > > [-] Limited evaluation. I think only testing on ScalarFlow is too limited to evaluate the effectiveness of this paper. The authors should include one more dataset to evaluate, even if it is a synthetic dataset.
> > > >
> > > > Limitation:
> > > > [+] The limitation mentioned they only consider gas. And in the rebuttal, the authors further explained the reason. Adding the explanation can improve the quality of this paper.
> > > >
> > > > In conclusion, my concerns are (1) the inaccurate statement of their interests, and (2) only testing on one dataset. The first concern can be easily solved. But the new experiment is hard to be conducted before the end of the rebuttal. After a comprehensive reconsideration of the manuscript and rebuttal, I tend to give a 'Borderline accept' as long as the author can clarify their interest in gas research and add the content of the rebuttal in the final version.

---

> > > > > ### Author Response · Authors · 2023-08-17
> > > > > **Thank you for your feedback and suggestions!**
> > > > >
> > > > > Dear Reviewer qUcR,
> > > > >
> > > > > Thank you for updating your reviews and providing your valuable feedback! Following your suggestions, we will incorporate all contents presented in this rebuttal (both the global response and the discussion in this thread) to our revised manuscript for better clarity.
> > > > >
> > > > > Sincerely,
> > > > >
> > > > > Authors of submission 1335

---

### Official Review · Reviewer_5YF8 · 2023-07-09

**Soundness:** 3 good
**Presentation:** 3 good
**Contribution:** 3 good
**Rating:** 5
**Confidence:** 4

**Summary:**

In this paper, the authors propose a method (HyFluid) to infer the fluid density and velocities from multiview videos. To deal with the visual ambiguities of fluid, physics-losses are introduced to try to enforce physics plausible velocities. A neural velocity field and a vortex particle-based velocity are introduced to capture the irrotational energy and model residual turbulent velocity respectively. Th experiments show that the proposed method has the ability to recover the vortical flow details to some extent.

**Strengths:**

The paper is overall well written and organized.
The proposed density loss and projection loss are well derived from the fluid mechanics.
The experiments show that the proposed method can perform better than the baselines.


**Weaknesses:**

The authors claim that the method can give physics plausible estimation of fluid fields. The physics intuitions for the density loss and projection are clear according to the incompressible condition. However, I don't see obvious physics intuition for the laminar loss.

For the problem that visual appearance of fluids depends on lighting and fluid substance properties, there is no further exploration in this direction. The problem becomes more important when the fluid is liquid.

**Questions:**

- Does the MGPCG applied to every simulation step to project the velocity during the training? How is the runtime cost for computing the projected velocity?
- Why to encourage high-density regions to have non-zero velocity (line 137)? What is the physics intuition here?
- As the visual appearance of fluid highly depends on the rendering process, does the results very sensitive to the parameters of the renderer?


**Limitations:**

yes

---

> ### Author Rebuttal · Authors · 2023-08-10
>
> Thank you for your time and comments! Please see our response below.
>
> - **Physical intuition on laminar loss**:
> Laminar flow does not manifest local density change, and thus inferring it from pure visual observations is challenging. We introduce this regularization term to account for the fact that even in constant-density fluid regions, there can still be laminar flow. Since this also depends on prior knowledge of the fluid to reconstruct, the laminar loss takes a flexible form: the hyper-parameter $\gamma$ models the prior belief of having laminar flow, e.g., when $\gamma=0$ it allows zero velocity even in high-density regions.
>
> - **Further exploration on visual appearance**:
> Following your suggestion, we allow learning spatially and temporally varying appearances. Please see the results and discussion in the "Spatially varying appearance" in the global response above.
>
> - **Rendering parameter**:
> We use volumetric rendering in our formulation (Eq. (3) in the main paper). The parameters include {near plane $t_n$, far plane $t_f$, number of samples for numerial integration $N$}, where $t_n$ and $t_f$ are determined by centering it at the scene center (which is the geometric center of camera principal rays) and empirically scale it. We find that as long as there are enough samples (in our case, more than 64), the numerical integration is stable and thus the results are not sensitive to these parameters.
>
> - **On MGPCG**:
> We use MGPCG in every training step to compute the projection loss. We implement MGPCG using Taichi [Hu2019] that allows GPU acceleration. We use three levels with resolution 128^3. On average, this costs around 100ms per step. Thanks to our efficient implementation, the whole training takes only ~9 hours on a A100 GPU, as noted in our supplementary material.
>
> **Reference**:
>
> [Hu2019] Hu, Y., Li, T. M., Anderson, L., Ragan-Kelley, J., & Durand, F. (2019). Taichi: a language for high-performance computation on spatially sparse data structures. ACM Transactions on Graphics (TOG), 38(6), 1-16.

---

> ### Author Response · Authors · 2023-08-14
> **Happy to answer any further questions**
>
> Dear Reviewer 5YF8,
>
> Thank you for reviewing our submission. We have posted our response per your suggestions and questions. We are happy to discuss with you and answer any further questions. We are looking forward to your feedback!
>
> Best,
>
> Authors of submission 1335

---

### Author Rebuttal · Authors · 2023-08-10

We thank all reviewers for their time and feedback. We clarify that since our goal is to reconstruct plausible fluid fields from real videos, **all experiments in our main paper are on real captured data, as specified in L201**. Please find our summary of major changes and response to some common questions below. We will incorporate these changes to our revised paper.

**Summary of major changes per reviewers' suggestions**:

1. [*qUcR*, *3dGa*, *4cV8*, *MRuw*] We add evaluations on synthetic data that provides groundtruth 3D fields.
2. [*qUcR*, *3dGa*, *4cV8*, *MRuw*] We add evaluations on different viscosity levels using synthetic data.
3. [*5YF8*, *MRuw*] We add ablation study on using spatiotemporally-varying appearance.
4. [*3dGa*, *4cV8*, *MRuw*] We add discussion and comparison to GlobTrans which is a SOTA non-learning-based fluid reconstruction method.
5. [*qUcR*, *3dGa*, *MRuw*] We add discussion and comparison to NeuroFluid which is a recent fluid dynamics learning method.
6. [*4cV8*] We improve velocity visualization using slice rendering.
7. [*MRuw*] We add more references and discussion to recent related work.


**[*qUcR*, *3dGa*, *4cV8*, *MRuw*] Evaluation on synthetic data**:

We include synthetic examples for evaluating 3D density and velocity fields. We use ScalarFlow synthetic dataset generation code [Eckert2019]. We generate five examples with different inflow source (randomized inflow area and density distribution) with higher viscosity and another five examples with lower viscosity. Since numerical viscosity is unavoidable, we simply use different simulation domain resolution to synthesize fluids with different viscosity levels. For the low viscosity group, we use 100x178x100. For the high viscosity group, we use 80x142x80.

We compare to the state-of-the-art neural fluid reconstruction method PINF [Chu2022] which has shown competitive results in 3D fluid fields reconstruction. Since the simulation groundtruth are up to a scale, we use scale-invariant RMSE to measure the performance. We only compute metrics where groundtruth density is greater than $0.1$ to rule out empty space (which is otherwise dominant) for clearer quantitative comparison. In particular, we consider volumetric density error (by querying density networks at the simulation grid points) to evaluate density prediction, and warp error (i.e., using velocity to advect density and comparing to GT density) to evaluate both density and velocity prediction. We also report novel view re-simulation results. From the table below, we see that ours outperforms PINF on both 3D fields and 2D rendering, similar to our main paper's observations. We show qualitative examples in Figure R1 and R2 in the PDF.

We show quantitative results in Table 1 below. We observe that ours outperforms PINF in all metrics. This is consistent with our observations on real data in our main paper.

Table 1: Evaluation on **synthetic** data.
|| Density error$\downarrow$| Warp error$\downarrow$ |PSNR$\uparrow$|SSIM$\uparrow$|LPIPS$\downarrow$|
|-|-|-|-|-|-|
| PINF | 4.95   | 4.88| 25.34 | 0.8641 | 0.1845
| Ours | **2.89**|  **3.29** |**27.93** |**0.8643**|**0.1259**|


**[*qUcR*, *3dGa*, *4cV8*, *MRuw*] Evaluation on different viscosity levels**

In addition to overall evaluation above, we also include separate evalution on the different viscosity levels. We show the results in Table R1 in the PDF. We see that ours outperforms the SOTA method PINF on all metrics except for high-viscosity SSIM, likely due to that the high-viscosity velocity fields are dominated by laminar flows which PINF tends to recover.


**[*qUcR*, *3dGa*, *MRuw*] Comparison to NeuroFluid**:

NeuroFluid [Guan2022] is a recent method on learning fluid dynamics. We use NeuroFluid official code and their released pretrained transition model, and train it on ScalarFlow real dataset following their instructions. NeuroFluid assumes known initial state while in real data there is no groundtruth initial state; thus, we use the SOTA fluid reconstruction method PINF [Chu2022] to reconstruct the first frame as the initial state. We include a comparison to NeuroFluid in Table R2 and Figure R3 in the PDF.

From the results we observe that NeuroFluid does not produce meaningful novel view synthesis, since it does not target at real fluid reconstruction.


**[*3dGa*, *4cV8*, *MRuw*] Comparison to GlobTrans**:

GlobTrans [Franz2021] is a global grid optimization-based method specifically designed for fluid re-simulation and reconstruction. We use the official code release and show an comparison on Table R2 and Figure R4 in the PDF.

From the results we observe that ours allows much better reconstruction fidelity reflected by higher PSNR and SSIM, yet GlobTrans yields better LPIPS. We note that GlobTrans assumes known lighting and uses a more sophisticated shading model. In contrast, ours does not assume known lighting and thus can be more general.


**[*5YF8*, *MRuw*] Spatiotemporally varying appearance**:

We add an ablation on predicting spatiotemporally-varying color to account for spatially-varying lighting. We show the novel view video synthesis (which mainly evaluates appearances) results in Table R3 and Figure R5 in the PDF.

We observe that this does not lead to significant differences on the ScalarFlow real dataset. This may be due to that the fluid is homogeneous in material and that the capturing environment is controlled. For more complex scenes with complex lighting, using spatially-varying color may help further.


**Reference**:

[Chu2022] Physics informed neural fields for smoke reconstruction with sparse data. TOG2022

[Eckert2019] ScalarFlow: a large-scale volumetric data set of real-world scalar transport flows for computer animation and machine learning. TOG2019

[Guan2022] Neurofluid: Fluid dynamics grounding with particle-driven neural radiance fields. ICML2022

[Franz2021] Global transport for fluid reconstruction with learned self-supervision. CVPR2021

---

### Decision · Program_Chairs · 2023-09-21

**Decision:**

Accept (poster)

**Comment:**

At the conclusion of the discussion period the reviewers gave this paper the following scores: 6,5,5,7,5, i.e. they all recommended acceptance.  The work compares visually with the PINF method of Chu et al., (2022) in Figure 5 and compares quantitatively to it in Table 1 as well as with the NeRFlow (Du et al., 2021) results.  However, after the discussion period it was noted that an Arxiv paper from the 10th April, 2023 entitled “Inferring Fluid Dynamics via Inverse Rendering” (Liu et al. 2023) [1], also submitted to ICLR 2023 and available on OpenReview has a number of similarities. To be specific, the overall framework of the method in this submission is quite similar to Liu et al. (2023), i.e. the integration of a fluid simulator with a differentiable renderer to construct a fully-differentiable system. Both use the learned/ optimized velocity to advect fluid density and both use a NeRF-like approach to render frame by taking the advected density field as input. It was also noted that an important contribution and element of novelty in this submission consists of inferring fluid fields from videos which was also a goal and contribution of Liu et al. (2023); however, other prior work has also examined the ScalarFlow smoke video dataset of Eckert et al., (2019), which mitigates this issue. However, some methodological details are similar or the same, such as the fact that both use an Eulerian simulator to alleviate visual ambiguities and divergence-free loss to keep fluid volume, which is likely discussed for the first time in Liu et al., (2023).  This point should really be mentioned in the submission.  Some experimental settings are also the same as in Liu et al. (2023). For example, the “scene composition” experiments in the submission is extremely similar to the “scene editing” experiments in Liu et al. (2023). This setup is used to prove that the general approach has a wide range of application scenarios. The contributions of Liu et al., (2023) relative to the contributions provided by this manuscript need to clarified. If this paper is modified to appropriately cite Liu et al., (2023) and place the contributions of the submission in appropriate context, then this paper is suitable for acceptance.

[1] Liu J, et al. Inferring Fluid Dynamics via Inverse Rendering. arXiv preprint arXiv:2304.04446, 2023.